# Glucagon-like Peptide-1 Receptor Agonists in Patients with Type 2 Diabetes Mellitus and Nonalcoholic Fatty Liver Disease—Current Background, Hopes, and Perspectives

**DOI:** 10.3390/metabo13050581

**Published:** 2023-04-23

**Authors:** Georgiana-Diana Cazac, Cristina-Mihaela Lăcătușu, Gabriela Ștefănescu, Cătălina Mihai, Elena-Daniela Grigorescu, Alina Onofriescu, Bogdan-Mircea Mihai

**Affiliations:** 1Unit of Diabetes, Nutrition, and Metabolic Diseases, Faculty of Medicine, “Grigore T. Popa” University of Medicine and Pharmacy, 700115 Iași, Romania; georgiana-diana.cazac@d.umfiasi.ro (G.-D.C.);; 2Clinical Center of Diabetes, Nutrition and Metabolic Diseases, “Sf. Spiridon” County Clinical Emergency Hospital, 700111 Iași, Romania; 3Unit of Medical Semiology and Gastroenterology, Faculty of Medicine, “Grigore T. Popa” University of Medicine and Pharmacy, 700115 Iasi, Romania; 4Institute of Gastroenterology and Hepatology, “Sf. Spiridon” County Clinical Emergency Hospital, 700111 Iași, Romania

**Keywords:** fatty liver, liver fibrosis, steatosis, inflammation

## Abstract

Nonalcoholic fatty liver disease (NAFLD) represents the most common chronic liver disease worldwide, reaching one of the highest prevalences in patients with type 2 diabetes mellitus (T2DM). For now, no specific pharmacologic therapies are approved to prevent or treat NAFLD. Glucagon-like peptide-1 receptor agonists (GLP-1RAs) are currently evaluated as potential candidates for NAFLD treatment in patients with T2DM. Some representatives of this class of antihyperglycemic agents emerged as potentially beneficial in patients with NAFLD after several research studies suggested they reduce hepatic steatosis, ameliorate lesions of nonalcoholic steatohepatitis (NASH), or delay the progression of fibrosis in this population. The aim of this review is to summarize the body of evidence supporting the effectiveness of GLP-1RA therapy in the management of T2DM complicated with NAFLD, describing the studies that evaluated the effects of these glucose-lowering agents in fatty liver disease and fibrosis, their possible mechanistic justification, current evidence-based recommendations, and the next steps to be developed in the field of pharmacological innovation.

## 1. Introduction

Type 2 diabetes mellitus (T2DM) is a chronic health condition that shares a wide range of risk factors and pathogenic elements with obesity and nonalcoholic fatty liver disease (NAFLD) [1]. The prevalence of diabetes in adults aged 20–79 years worldwide is estimated at 10.5%, of whom more than 90% have T2DM [2].

Along with the diabetes pandemic, today, NAFLD represents the most prevalent liver disease worldwide and is perceived as the hepatic manifestation of the metabolic syndrome [3]. The global prevalence of NAFLD among adults is currently 25–30% [4] and is predicted to reach 55.7% in 2040 [5]. More than 90% of the patients with severe obesity, and 50% of T2DM patients have NAFLD [4,6]. T2DM is often associated with multiple organ dysfunction, including eye, kidney, liver, and cardiovascular disease. NAFLD represents the least screened diabetes mellitus complication today; this is a discouraging finding, given the rapid progression towards nonalcoholic steatohepatitis (NASH), fibrosis, cirrhosis, or even hepatocellular carcinoma, all leading to a rising liver-related mortality [7].

The relationship between NAFLD and T2DM is bidirectional and requires increased awareness [8]. The presence of diabetes promotes the progression of NAFLD toward advanced liver disease, while the presence of NAFLD increases the risk of developing other diabetes-related complications or even worsening the already present ones [9,10].

Accumulating evidence supports the use of some antidiabetic agents, such as pioglitazone and glucagon-like peptide-1 receptor agonists (GLP-1RAs), to improve histological findings of NASH and fibrosis in T2DM patients [11]. Well-designed trials with these agents have been published, but the insufficient duration of the studies for both these classes limits the significance of their results [12]. In the case of GLP-1RAs, the hepatic benefits found in several studies that focused on NAFLD-associating patients have added to the already acknowledged effects of improving cardiovascular outcomes, preventing the nephropathy progression and reducing all-cause mortality in T2DM patients [13].

GLP-1RAs can be an effective treatment in patients with T2DM and NAFLD, so the aim of this comprehensive review is to summarize the existing evidence on the effectiveness of GLP-1RA therapy in the management of NAFLD that complicates T2DM, describing the studies that evaluated the effects of these glucose-lowering agents in fatty liver disease and fibrosis, their potential mechanistic justification, current evidence-based recommendations, and the next steps to be developed in the field of pharmacological innovation.

## 2. Basic Concepts in the Management of Patients with T2DM and NAFLD

Screening of people at risk for liver disease is advised for all patients with T2DM, as well as in individuals who are overweight or living with obesity, in people with metabolic syndrome, or in those living with cardiovascular disease [14,15].

Once the diagnosis of NAFLD is established, an intensive management strategy needs to be implemented. The cornerstone of NAFLD treatment remains lifestyle optimization; paradoxically enough, given the increased incidence of the disease and the vital risks involved by its progression, there are still no drugs specifically approved for NAFLD therapy [14,16]. The treatment objectives involve avoidance or delayed progression of diabetes-related complications (cardiovascular disease and liver cancer included) by improving the metabolic profile, and the reduction of the liver fat load, inflammation, or even fibrosis, given the liver’s ability to regenerate [14].

Lifestyle modification in the management of NAFLD includes initiation and maintenance of weight loss through nutritional intervention and increased physical activity patterns [17,18]. Nevertheless, nonpharmacological therapy displays limited efficacy for weight-related outcomes, which impairs its ability to change the course of liver disease [19].

Antihyperglycemic agents supporting weight loss in patients with T2DM have shown beneficial effects in NAFLD [20]. However, it appears that improvement of glycemic control in patients with co-existent NAFLD and diabetes may be associated with an improvement in NAFLD independently of the lost weight [21]. Glycemic outcomes cannot be considered outside the need to tackle metabolic traits such as dyslipidemia or obesity, which improves the liver condition in patients with or without diabetes [22].

Along with sodium-glucose cotransporter 2 (SGLT2) inhibitors, GLP-1RAs have become the first-line therapy for T2DM due to their potent glucose-lowering effects and their ability to reduce the cardiovascular risk irrespective of the glycemic improvement or the weight loss [23].

GLP-1RAs belong to the incretin-based therapies with subcutaneous or oral administration that are currently approved for the treatment of T2DM and obesity (Table 1) [24,25]. They are intensively scrutinized, given their beneficial impact on the glycemic control, their ability to reduce the visceral adipose tissue and the total body weight, and their cardio-reno-metabolic benefits in patients with T2DM and obesity (Figure 1) [26,27,28]. Therefore, the inference of their benefits in NAFLD-complicated diabetes, or even in NAFLD without associated diabetes, was only a step further [29]. Some data even suggest that GLP-1RAs have metabolic effects related to NAFLD pathophysiology [22].

Some of the high-impact guidelines for NAFLD have already embedded the use of GLP-1RAs (Table 2). United States guidelines were the first to support the use of GLP-1RAs in patients with T2DM and NASH [12,15]. As of now, further studies are still needed to clarify the role of GLP-1RAs as an NAFLD-targeted therapy in persons with or without diabetes, including their ability to interfere with each of the histological changes seen in this metabolic liver disease, as a valuable addition to their well-recognized capacity to improve cardiometabolic health [39].

## 3. Search Methods

We performed an extensive literature search of PubMed and ClinicalTrials.Gov databases using the terms “GLP-1 receptor agonist” or “GLP-1RA” combined with “fatty liver” or “NAFLD” or “NASH” or “fibrosis” and “diabetes mellitus” or “DM” or “diabetic patients” or “T2DM”. Further research was made using the generic names of the GLP-1RA representatives, specifically “exenatide”, “lixisenatide”, “dulaglutide”, “liraglutide”, “semaglutide”, “albiglutide”, and “efpeglenatide” for trials that studied their effect on NAFLD in patients with T2DM at any time up to February 2023.

Clinical trials, randomized controlled trials (RCT), and multicenter studies examining the effectiveness of GLP-1RA therapy on NAFLD in patients with T2DM were included. Studies that did not include subjects with T2DM were excluded. Only articles available in English and in full text were considered. To avoid omissions, other types of articles such as reviews, guidelines, systematic reviews, and meta-analyses were consulted. We also carefully examined the reference lists of the previously selected articles for other eligible and relevant articles. All the relevant information derived from the selected articles has been compiled into text or table form.

Relevant data were extracted by two independent authors who selected information related to the study characteristics (author, year of publication, country of origin, sample size, follow-up duration), the therapeutic regimen, diagnostic methods, and treatment outcomes (changes in liver biomarkers, steatosis and fibrosis indices, liver fat content, and histological modifications). An independent third author performed blinded checking of the data and disparities were resolved through consensus.

## 4. Search Results

Table 3 summarizes all data (clinical trials, RCT, multicenter studies, post hoc analyses) with at least a 12-week follow-up collected from patients with T2DM and NAFLD based on the use of exenatide, liraglutide, dulaglutide, and semaglutide in various doses compared with either placebo or other glucose-lowering agents, regardless of the population sample, primary endpoints, or investigation methods. No studies with efpeglenatide, lixisenatide, or albiglutide were found. All cited references summarized results of human studies, with both male and female adult patients included. Patients in various studies had different degrees of NAFLD, from steatosis to NASH or fibrosis. A total of 32 studies were found, including eight studies with exenatide, sixteen with liraglutide, six studies with semaglutide, and two studies with dulaglutide, all with positive results of improving NAFLD. Among all, liraglutide was by far the most investigated GLP-1RA for the management of NAFLD in patients with T2DM. In addition to the effects of ameliorating liver biomarkers, steatosis and fibrosis indices, liver fat content, and histological data, some studies also correlated the amounts of visceral adipose tissue and subcutaneous adipose tissue to NAFLD, and found improvements after the treatment with GLP-1RAs was implemented.

### 4.1. Effects of GLP-1RAs on Diabetes-Related Liver Disease

#### 4.1.1. Changes in Serum Liver Enzymes

Exenatide was the first GLP-1RA to show, besides its favorable effects on HbA1c, insulin resistance, body weight, and blood pressure, an improvement of the hepatic enzymes in patients with T2DM [45,49]. Nowadays, exenatide is less used because of the lack of cardiorenal benefits [74] that were later demonstrated by the new generation of long-acting GLP-1RAs, such as liraglutide, dulaglutide, and semaglutide [75].

Liraglutide appears to be, for now, the most studied GLP-1RA in NAFLD, raising high hopes for its ability to prevent and treat the metabolic liver disease [76,77]. A meta-analysis of the LEAD program that included 4442 patients with T2DM assessed the effects of liraglutide on liver parameters after a 26-week therapy [78]. Liraglutide 1.8 mg per day improved liver enzymes, while smaller doses did not show significant effects after adjustments to body weight were applied [78].

The findings of a post hoc analysis of 1499 subjects from the AWARD trials with dulaglutide also showed a decrease in serum aminotransferases vs. placebo (−8.8 IU/L vs. −6.7 IU/L) in patients with T2DM and NASH after the administration of dulaglutide 1.5 mg per week [79].

Semaglutide is another GLP-1RA that has also gained interest in the treatment of steatohepatitis, as several studies in patients with T2DM and obesity reported beneficial changes in liver enzymes and inflammatory biomarkers such as high-sensitivity C-reactive protein [80,81].

An earlier meta-analysis of patients with T2DM and a high risk of NAFLD concluded that lixisenatide might have beneficial effects on liver enzymes compared to the control group [82].

#### 4.1.2. Effects on Composite Indices of Hepatic Steatosis and Fibrosis

Improvement of noninvasive scores for NAFLD was observed in a trial by Gastaldelli et al., which assessed the efficacy of exenatide once weekly in association with dapagliflozin in T2DM patients. Fatty liver index (FLI), NAFLD liver fat score (NLFS), Fibrosis-4 (FIB-4) index, and NAFLD fibrosis score (NFS) decreased under the synergic effects of exenatide and dapagliflozin [42]. On the other hand, multiple glucose-lowering agents were able to improve NAFLD scores in patients with T2DM, independently of their ability to induce weight loss. In a retrospective study, Colosimo et al. demonstrated a correlation between the glycated hemoglobin level, which was used to identify good glycemic responders, and the changes in FLI and FIB-4, independently of the body mass index (BMI) or the nature of the antidiabetic medication, represented by either GLP-1RAs, dipeptidyl peptidase-4 (DPP-4) inhibitors, or SGLT2 inhibitors [21].

Gameil et al. performed a study that compared the effect of liraglutide and dulaglutide vs. conventional treatment on FLI and FIB-4 index in patients with T2DM complicated with NAFLD. Liraglutide and dulaglutide led to significant improvements in the FLI and FIB-4 values, with a greater change in the liraglutide group [52].

#### 4.1.3. Changes in Liver Fat Content or Fibrosis Evaluated by Imaging Techniques

Several studies investigated, using imaging techniques, the GLP-1RAs effect on the reduction of liver fat content as a primary outcome in patients with T2DM and NAFLD. The reduction of liver fat content was assessed by either ultrasound [45,56,59,69] or magnetic-resonance-based imaging methods [43,64,72]. Liraglutide improved the intrahepatic fat content, leading to a decrease of approximately 60% from baseline when compared with metformin and gliclazide, along with reducing mean HbA1c levels from 8.9% to 5.9% and transaminase levels [59]. Another study with liraglutide pointed to a close correlation between the degree of hepatic steatosis evaluated by ultrasound and the extent of weight loss [60]. Moreover, other authors suggest that the diminution of liver steatosis due to the significant decreases in body weight and HbA1c values can also lead to a reduction of the hepatic inflammation [83].

However, mixed results are reported in this category. Smits et al. did not observe any valuable effect after a 12-week liraglutide or sitagliptin administration in T2DM, where proton magnetic resonance spectroscopy (^1^H-MRS) was used to measure hepatic fat and NFS, FIB-4 index, and AST to platelet ratio index (APRI) scores to assess fibrosis [62]. Another early study on T2DM patients randomized to insulin glargine or liraglutide for 12 weeks showed no significant change in liver proton density fat fraction measured by MRS [64], while dulaglutide was also not able to demonstrate a reduction of the intrahepatic fat estimated by transient elastography in NAFLD patients, probably due to the short duration of follow-up [84].

Contrariwise, Flint et al. found that semaglutide can determine more than 30% reduction in hepatic steatosis vs. placebo; however, no significant improvement of the liver stiffness assessed by magnetic resonance elastography (MRE) and of the magnetic resonance imaging-estimated proton density fat fraction (MRI-PDFF) was seen [70].

In short, the majority of studies acknowledge that GLP-1RAs decrease or even normalize the serum liver enzyme levels and reduce the liver fat content on imaging in people with NAFLD and T2DM [29,76].

#### 4.1.4. Effects on Biopsy-Proven Histopathological Modifications

Liver biopsy and subsequent histologic examination represent the “gold standard” in diagnosing NAFLD [11]. There are only two important RCTs that have assessed the evolution of liver biopsy-proven histopathological abnormalities in diabetic and nondiabetic subjects with NASH [29,85]. The 48-week LEAN trial led by Armstrong and colleagues showed that liraglutide improved some features of NASH by delaying fibrosis progression when compared to placebo (39% vs. 9%, *p* = 0.019). However, the NAFLD Activity Score (NAS) did not display any significant changes during the trial [61]. The other RCT compared various doses of an atypical administration of once-daily subcutaneous semaglutide versus placebo for 72 weeks in patients with or without T2DM with histologic evidence of NASH. The study concluded that NASH resolution was dose-dependent, seen in 59% of those in the 0.4 mg/day semaglutide group (equivalent to 2.4 mg/week) versus 17% in the placebo group (*p* < 0.001) and due to the significant weight loss [71]. The treatment group exhibited a slower progression of fibrosis (4.9% vs. 18.8% in the placebo group), but no significant improvement in the severity of fibrosis [71]. Therefore, liraglutide and semaglutide achieved the histological resolution of NASH in 40 to 60% of participants, but with no influence on the overall fibrosis outcome [81,86]. The Effect of Semaglutide in Subjects with Noncirrhotic Nonalcoholic Steatohepatitis (ESSENCE) study is an ongoing phase 3 larger trial assessing the effects of semaglutide in NASH-related fibrosis [15,87].

#### 4.1.5. Effects Assessed by Combined Investigations

Simeone et al. randomized patients with newly diagnosed T2DM and prediabetes to liraglutide 1.8 mg/dose or lifestyle counseling until achieving a target of approximately 7% less than the initial body weight. Both groups displayed significant comparable weight loss, improvement of the glycemic values, and reductions of the BMI, IL-1β level, and NAFLD degree assessed by magnetic resonance [54].

A 24-week, oral administration of semaglutide study assessed its efficacy and safety in T2DM complicated with NAFLD. A greater improvement was observed in the impaired liver function tests, hypertriglyceridemia, insulin resistance, and hepatic steatosis, led by the favorable effects on glycemic control and body weight. Controlled attenuation parameter (CAP) values significantly decreased from 344 to 279 dB/m at 24 weeks, the FIB-4 index decreased from 1.42 to 1.1, ferritin decreased from 4.1 ng/mL to 3.5 ng/mL, and hepatic enzymes were normalized at 24 weeks. However, liver stiffness measurement (LSM) values showed no significant changes in fibrosis after the 24 weeks of semaglutide treatment [67].

A systematic review and meta-analysis of 26 studies by Kumar et al. found that GLP-1RAs are associated with a significant reduction in ALT (mean difference (MD) −27.98, *p* = 0.04) and GGT (MD −40.65, *p* = 0.03), with no statistically significant influence upon AST values. They also demonstrated a significant improvement in liver steatosis (standard MD −2.53, *p* = 0.03), with insufficient data to support an effect on inflammation and fibrosis [88].

According to a meta-analysis of seven multinational RCTs assessing NAFLD by biopsy or imaging techniques, exenatide, and liraglutide proved to have consistent beneficial effects on serum liver enzymes, liver fat content, intraabdominal adipose tissue, and histological resolution of NASH [89].

At present, more large RCTs with biopsy-assessed outcomes are needed to accurately evaluate the effects of GLP-1RAs on various histopathological findings of NAFLD in patients with or without diabetes [77,83].

## 5. Potential Beneficial Mechanisms Underlying the Effects of GLP-1RAs in Diabetes-Related Liver Disease

### 5.1. Indirect Effects That Promote NAFLD Improvement

The mechanisms involved in NAFLD pathophysiology are not completely understood. Current data link the development of fatty liver disease to insulin resistance, de novo lipogenesis, oxidative stress, microbiome dysregulation, immune or cytokine-related anomalies, mitochondrial injury, and apoptosis, overlapping a pre-existing genetic predisposition [7,90]. This multifactorial determinism suggests that NAFLD therapy should be as personalized as possible to optimally interfere with its predominant underlying mechanisms in each individual.

GLP-1RAs appear to have pleiotropic effects that can improve the metabolic liver disease in patients with T2DM [91]. Their effects on NAFLD may be due to systemic benefits upon various metabolic pathways, and not to the activation of the local GLP-1 receptors, which are not expressed in the hepatic tissue in significant amounts [92]. Among such metabolic effects, the GLP-1RA class mainly includes the stimulation of insulin release and the inhibition of glucagon secretion, a decreased hepatic glucose production, an increased insulin sensitivity in hepatocytes and adipocytes, a delayed gastric emptying, and the suppression of appetite [61,93].

This perspective is supported by the results of earlier studies, which indicated that an intensive glycemic control may ameliorate liver fibrosis seen on serial hepatic biopsies in patients with T2DM [94].

The loss of body weight by appetite suppression and inducing satiety is one of the main intermediate steps leading to the beneficial effects of GLP-1RAs. One example is the Lira-NAFLD study, where liraglutide 1.2 mg per day induced a mean weight loss of 3.6 kg and a proportional decrease of the liver fat content from 17.3% to 11.9% [60].

GLP-1RAs also have antioxidant and anti-inflammatory properties leading to a significant diminution of the oxidative stress and inflammatory biomarkers, which could determine NASH amelioration and some protection against the natural evolution of the liver fibrosis [95].

The gut microbiome plays an important role in the occurrence and development of NAFLD and T2DM, and can therefore emerge as a forthcoming target in the therapy of these metabolic diseases [96,97]. Animal studies provide some evidence that liraglutide has a beneficial impact on components of the intestinal flora that are related to inflammation and glucolipid metabolism, thus improving the fatty liver disease [98]. Liraglutide also seems to have a beneficial effect on the human intestinal microbiome in studies on patients with NAFLD, where it decreased the inflammatory factors and improved the liver function and the adipose content [99].

### 5.2. Direct Mechanisms That Promote NAFLD Improvement

Another pathway that warrants future investigation is the involvement of glycerophospholipid perturbance in T2DM pathogenesis [100]. Du et al. tried to use the changes in metabolites after treatment with two GLP-1RAs (liraglutide and dulaglutide) to evaluate the metabolic remodeling effects [101]. After assessing various metabolites in patients with T2DM, many studies pointed out that some of them contribute to specific metabolic dysfunctions, such as insulin resistance, obesity, glucose impairment, and NAFLD. At present, the most preeminent class of T2DM-associated metabolites is represented by glycerophospholipids. The authors of the abovementioned study point out that, for now, there is no clear understanding yet of how GLP-1RAs reduce major adverse cardiovascular events (MACE) or how they ameliorate NASH, as these effects are not fully explained by the indirect beneficial effects on weight, lipid profile, or blood pressure. They hypothesize that such effects may be determined by the effects liraglutide and dulaglutide have on different metabolic pathways, among which the alteration of glycerophospholipid metabolism may play an important role in the pathophysiology of NAFLD [101].

Another study investigated the effect of liraglutide on thyroid resistance in the liver of T2DM patients [102]. Relevant findings suggest that the intrahepatic impairment of the thyroid hormone (TH) pathway due to thyroid hormone receptor beta (THRB) gene mutations may lead to fatty liver; TH resistance can thus take part in NAFLD development [29]. In the abovementioned study, liraglutide decreased thyroid hormone levels in the NAFLD group of patients with T2DM. The results suggest that a mechanism by which liraglutide can improve hepatic TH resistance by restoring the impaired THRB expression is likely possible [102].

## 6. Perspectives

The twincretin tirzepatide is a novel dual glucagon-like peptide-1 (GLP-1) and glucose-dependent insulinotropic polypeptide (GIP) receptor agonist approved by the FDA in May 2022 for the treatment of T2DM, based on the results of the Study of Tirzepatide in Participants with T2DM Not Controlled with Diet and Exercise Alone (SURPASS) clinical trials [103,104]; benefits in patients with obesity were also proved later [105]. A randomized trial of patients with T2DM who received tirzepatide revealed significant decreases in biomarkers related to NASH such as AST/ALT, procollagen III, keratin 18, and an increase in adiponectin levels, which is believed to activate antifibrotic and antisteatogenic effects in the liver [106]. A once-weekly subcutaneous administration of tirzepatide for 72 weeks in obese patients without diabetes demonstrated weight losses of 15% with the 5 mg dose, 19.5% with the 10 mg dose, and 20.9% with the 15 mg dose vs. a 3.1% weight loss in the placebo group. This RCT also showed changes in the fasting insulin levels, lipid profile, and blood pressure, along with a specific reduction of the fat mass [105]. This may be the primary indirect pathway leading to NAFLD improvement, but additional investigations are needed. Moreover, the prediabetes group that received tirzepatide had regressed to normoglycemia [105]. Given these beneficial effects in the metabolic profile and intrahepatic fat, an RCT with tirzepatide in biopsy-proven NASH individuals with or without T2DM is currently ongoing and will complete its results in 2024 [107].

Cotadutide is a dual GLP-1/glucagon (Gcg) receptor dual-agonist investigated at present for T2DM, kidney disease, heart failure, and NASH [108]. Cotadutide exerts a unique dual agonism mode of action upon the glucose and lipid metabolism that appears to be superior compared with GLP-1 agonism alone, given the indirect effects of reduced inflammation and reversed fibrosis found in preclinical models, in addition to significant properties to induce HbA1c and weight reduction and to improve insulin resistance [109]. In a 54-week phase 2b study, cotadutide led to positive changes in body weight, serum ALT levels, and scores of hepatic steatosis and fibrosis (NFS and FIB-4) in patients with NASH and T2DM [110].

Another relevant trial is the one-year administration of liraglutide to 82 women with NAFLD and history of gestational diabetes, followed by a major reduction of steatosis vs. placebo [111].

Last, but not least, it must be mentioned that multiple ongoing or forthcoming large trials are aiming to investigate, using liver biopsy or MRI techniques, the effects of GLP-1RAs and their newly derived molecules on patients with T2DM and NAFLD (Table 4). Their results will no doubt shape the clinical guidelines for pharmacological therapy in T2DM and NAFLD patients.

The increased appearance of NAFLD in patients with T2DM in a large proportion of patients, added to the still-incomplete knowledge of the mechanical changes that these therapies induce in the liver, supports the need for more studies that seek effective treatments for both pathologies. A new contingent of drugs under development based on the principle of GLP-1 receptor agonism may represent a turning point in this field in the coming years.

## 7. Conclusions

Given the increasing worldwide incidence of both T2DM and NAFLD, this comprehensive review shows that in type 2 diabetic patients, GLP-1RAs may contribute to the reduction of liver fat content, induce NASH resolution, and reduce the progression of fibrosis.

## Figures and Tables

**Figure 1 metabolites-13-00581-f001:**
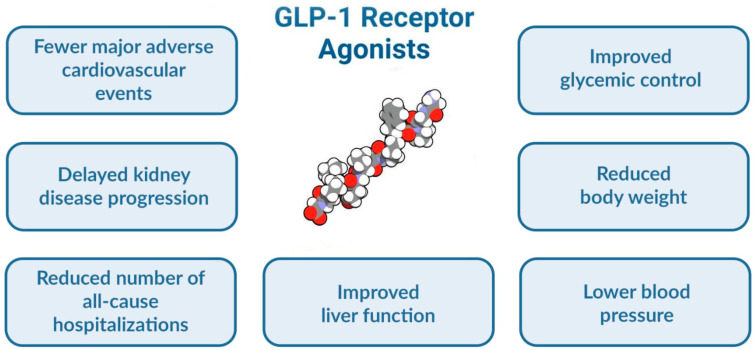
The beneficial effects of GLP-1RAs on various cardiometabolic components associated with T2DM.

**Table 1 metabolites-13-00581-t001:** Characteristics of GLP-1RAs approved for the treatment of T2DM [24,30].

GLP-1RA Generic Name, Ref.	Trade Name Manufacturer	First Approved	Dose and Route of Administration	Administration Schedule	Side Effects
Short-acting					Nausea, vomiting, diarrhea. Pruritus and erythema in the injection area of administration. Risk of pancreatitis and pancreatic cancer. Risk of gallbladder and biliary disease [31].
Exenatide [32]	Byetta^®^ AstraZeneca, Cambridge, England	2005 (US) 2006 (EU)	5–10 mcg sc	Twice daily prior to meals	
Lixisenatide [33]	Adlyxin^®^, Lyxumia^®^ Sanofi, Paris, France	2013 (EU) 2016 (US)	10–20 mg sc	Once daily prior to the first meal	
Long-acting					
Exenatide ER [32]	Bydureon^®^ AstraZeneca, Cambridge, England	2012 (US) 2011 (EU)	2 mg sc	Once weekly, unrelated to meals	
Albiglutide [34]	Eperzan^®^ Tanzeum^®^ GlaxoSmithKline, London, England	2014 (EU) 2014 (US)	30–50 mg sc	Once weekly, unrelated to meals	
Liraglutide [35]	Victoza^®^ Novo Nordisk, Bagsvaerd, Denmark	2010 (US) 2009 (EU)	0.6–1.8 mg sc	Once daily, unrelated to meals	
Dulaglutide [36]	Trulicity^®^ Eli Lilly and Company, Indianapolis, Indiana, USA	2014 (US, EU)	0.75–1.5 mg sc	Once weekly, unrelated to meals	
Semaglutide [37,38]	Ozempic^®^ Novo Nordisk, Bagsvaerd, Denmark	2017 (US) 2018 (EU)	0.25–1 mg sc	Once weekly, unrelated to meals	
	Rybelsus^®^ Novo Nordisk, Bagsvaerd, Denmark	2019 (US) 2020 (EU)	3–14 mg po	Once daily, one hour prior to the first meal	

Abbreviations: GLP-1RAs, glucagon-like peptide-1 receptor agonists; Ref., reference; US, United States; EU, Europe; sc, subcutaneously; po, oral.

**Table 2 metabolites-13-00581-t002:** The place of GLP-1RAs among various guidelines as a clinical approach to NAFLD in T2DM patients.

Guidelines/Guidance	Recommendations
AASLD Practice Guidance on the Clinical Assessment and Management of Nonalcoholic Fatty Liver Disease, 2023 [15]	Semaglutide can be considered for T2DM or obesity in patients with NASH, adding cardiovascular benefit and improving NASH.
AACE Clinical Practice Guideline for the Diagnosis and Management of Nonalcoholic Fatty Liver Disease in Primary Care and Endocrinology Clinical Settings, 2022 [12]	Recommend pioglitazone and GLP-1RAs for people with T2DM and biopsy-proved NASH. Consider treating T2DM with GLP-1RAs or pioglitazone in the situation of possible NASH with modified noninvasive tests and elevated levels of hepatic enzymes, also offering cardiometabolic benefits, even in pediatric obesity and T2DM.
APASL clinical practice guidelines for the diagnosis and management of metabolic-associated fatty liver disease, 2020 [40]	Insufficient evidence in the Asian population.
EASL–EASD–EASO Clinical Practice Guidelines for the management of nonalcoholic fatty liver disease, 2016 [17]	Pharmacotherapy reserved for patients with NASH and significant fibrosis (≥stage F2) or NASH with a high risk for disease progression (elevated ALT, T2DM, MS). No firm recommendation for the use of pioglitazone or vitamin E in NASH. Insufficient evidence for GLP-1RAs.
NICE Guideline on Liver Disease (NAFLD), 2016 [41]	GLP-1RAs not mentioned. Insufficient evidence for pioglitazone and vitamin E.

Abbreviations: AASLD, American Association for the Study of Liver Diseases; AACE, American Association of Clinical Endocrinology; APASL, The Asian Pacific Association for the Study of the Liver; EASL, European Association for the Study of the Liver; EASD, European Association for the Study of Diabetes; EASO, European Association for the Study of Obesity; NICE, National Institute for Health and Care Excellence; NASH, nonalcoholic steatohepatitis, GLP-1RA, glucagon-like peptide-1 receptor agonist; T2DM, type 2 diabetes mellitus; MS, metabolic syndrome; ALT, alanine aminotransferase.

**Table 3 metabolites-13-00581-t003:** Summary of studies that evaluated the effects of GLP-1RAs in patients with T2DM and NAFLD.

GLP-1RA	Author, Year, Ref.	Country	Total Patients (GLP-1RA/Control)	Control Arm	GLP-1RA Arm	Follow-Up Duration	Diagnosis Method	Results of GLP-1RAs
Exenatide	Gastaldelli et al., 2020 [42]	Italy	228/227/230	Dapagliflozin + placebo Exenatide + placebo	Exenatide 2 mg OW and Dapagliflozin 10 mg daily	104 weeks	Steatosis and fibrosis scores	Improved biomarkers of steatosis and fibrosis in the exenatide and dapagliflozin group vs. exenatide or dapagliflozin alone.
	Liu et al., 2020 [43]	China	35/36	Glargine insulin	Exenatide 5 μg BID/4 weeks, then 10 μg BID/20 weeks	24 weeks	^1^H-MRS	Greater reductions in body weight, visceral fat area, liver enzymes, FIB-4, postprandial plasma glucose, and LDL-C.
	Dutour et al., 2016 [44]	France	22/22	Standard of care without GLP-1RA	Exenatide 5 μg BID/4 weeks, then 10 μg BID/22 weeks	26 weeks	^1^H-MRS	Reduced liver fat content and epicardial fat dependent on weight loss.
	Shao et al., 2014 [45]	China	30/30	Insulin-based therapy	Exenatide 5 μg BID/4 weeks, then 10 μg BID/8 weeks	12 weeks	US	Lower AST, ALT, and GGT (*p* < 0.001) correlated with body weight change.
	Fan et al., 2013 [46]	China	49/68	Metformin	Exenatide 5 μg BID/4 weeks, then 10 μg BID/8 weeks	12 weeks	US	Controlled blood glucose, reduced body weight, and improved hepatic enzymes.
	Cuthbertson et al., 2012 [47]	United Kingdom	25 19/6	-	Exenatide 5 μg BID/4 weeks, then 10 μg BID/20 weeks Or Liraglutide 0.6–1.2 mg daily	24 weeks	^1^H-MRS MRI	Correlation between intrahepatic lipid and AST, ALT, and GGT levels (*p* < 0.05). Significant body weight relative reduction of 4.3%. Reduction in VAT and SAT volume.
	Sathyanarayana et al., 2011 [48]	USA	11/10	Pioglitazone	Exenatide 5 μg BID/2 weeks, then 10 μg BID/50 weeks	52 weeks	MRS, plasma adiponectin	Significantly greater reduction in hepatic fat (Δ = 61% vs. 41%, *p* < 0.05), and greater increase in circulating adiponectin in combination therapy vs. pioglitazone alone.
	Buse, 2007 [49]	USA	283	Placebo	Exenatide 5 μg or 10 μg BID	2 years	Liver enzymes	HbA1c and ALT reduction, insulin resistance improvement, weight loss, and blood pressure.
Liraglutide	Tan et al., 2022 [50]	China	262/1503	Conventional drug therapy	Liraglutide daily	12 months	Fibrosis scores LSM	Decrease of NFS (*p* = 0.043), FIB-4 (*p* = 0.044), and LSM (*p* = 0.007). Reduced prevalence of advanced fibrosis (3.1% vs. 6.1%, *p* = 0.218) in the liraglutide group vs. control group.
	Bizino et al., 2020 [51]	Netherlands	23/26	Placebo	Liraglutide 1.8 mg daily	26 weeks	MRI	Significantly reduced SAT No significant change in VAT, hepatic, epicardial, and myocardial fat content.
	Gameil et al., 2020 [52]	Egypt	79/80/65	Conventional treatment	Liraglutide 1.8 mg daily Or Dulaglutide 1.5 mg OW	24 weeks	FLI FIB-4 score	Significant reduction of median FLI and FIB-4 score in the liraglutide and dulaglutide groups vs. the conventional treatment group (*p* < 0.001). Greater change in the liraglutide group vs. dulaglutide group (*p* = 0.027).
	Guo et al., 2020 [53]	China	32/32/32	Insulin glargine Placebo	Liraglutide 1.8 mg daily	26 weeks	^1^H-MRS	Significantly decreased IHCL (26.4% ± 3.2% to 20.6% ± 3.9%, *p* < 0.05). Significantly decreased of SAT and VATgroup (SAT, 331.7 ± 79.0 cm^2^ to 295.3 ± 80.3 cm^2^, *p* < 0.05; and VAT, 235.6 ± 30.8 cm^2^ to 188.2 ± 26.6 cm^2^, *p* < 0.05).
	Simeone et al., 2020 [54]	Italy	16 people with T2DM, 16 with prediabetes	Lifestyle counseling	Liraglutide 1.8 mg daily	Not mentioned	MRI	Improvement in glycemic control, CRP, IL-1β level, BMI, and NAFLD degree.
	Yan et al., 2019 [55]	China	24/27	Sitagliptin Glargine	Liraglutide 1.8 mg daily	26 weeks	MRI-PDFF	Reduced body weight, intrahepatic lipid, and VAT in addition to improving glycemic control.
	Tian et al., 2018 [56]	China	52/75	Metformin 1–1.5 g/day	Liraglutide 0.6–1.8 mg daily	12 weeks	B-mode Ultrasonic Scanning	Significant decrease of 2 h plasma glucose, AST, ALT, and adiponectin levels.
	Zhang et al., 2018 [57]	China	424/411	Conventional drug therapy	Liraglutide 1.2 mg daily	24 weeks	Biochemical analyzer	Significant improvement in blood glucose level, HbA1c, lipid profile, and liver function.
	Bouchi et al., 2017 [58]	Japan	8/9	Insulin	Liraglutide 0.9 mg daily	36 weeks	CT	Significantly reduced VFA, LAI, ACR, and CRP levels.
	Feng et al., 2017 [59]	China	29/29	Metformin Gliclazide	Liraglutide daily	24 weeks	US	Lower HbA1c, improved liver enzymes, weight loss on liraglutide and metformin. Decreased liver fat content in all groups, greater in liraglutide compared with others.
	Petit et al., 2017 [60]	France	68	Conventional drug therapy	Liraglutide 1.2 mg daily	24 weeks	MRS	Significantly reduced LFC (−31%, *p* = 0.0001) and body weight.
	Armstrong et al., 2016 [61]	United Kingdom	26/26	Placebo	Liraglutide 1.8 mg daily	48 weeks (extension to 72 weeks)	Liver biopsy	Significantly higher NASH resolution (39% liraglutide vs. 9% placebo). No worsening of fibrosis.
	Smits et al., 2016 [62]	Netherlands	17/17	Sitagliptin 100 mg Placebo	Liraglutide 1.8 mg daily	12 weeks	^1^H-MRS Fibrosis scores	Reduced glycemia, HbA1c, insulin levels. Reduced steatosis by 10% (*p* = 0.98). No influence on hepatic fibrosis. Reduced plasma albumin levels (*p* = 0.03).
	Vanderheiden et al., 2016 [63]	USA	35/36	Placebo	Liraglutide 1.8 mg daily	24 weeks	MRI/MRS	Significant reduction in abdominal SAT, no change in VAT, and the ratio of visceral to total fat.
	Tang et al., 2015 [64]	Canada	18/17	Glargine	Liraglutide 1.8 mg daily	12 weeks	MRI-PDFF	Similar improvement in the liver fat fraction in both groups (*p* = 0.34). No weight gain from insulin therapy.
	Ohki et al., 2012 [65]	Japan	26/36 26/20	Sitagliptin Pioglitazone	Liraglutide daily 0.3 mg/first week, 0.6 mg/next week, and finally up to the limit dose of 0.9 mg if necessary	48 weeks for liraglutide	US	Body weight decreased from 81.1 to 78 kg. BMI from 30.1 to 28.6 kg/m^2^. Fasting glycemia from 207 to 168 mg/dL. HbA1c from 8.4% to 7.6%. AST from 50 IU/L to 35 IU/L. ALT from 65 IU/L to 48 IU/L. APRI index from 0.73 to 0.49.
Semaglutide	Carretero-Gomez et al., 2023 [66]	Spain	213	-	Semaglutide sc OW	24 weeks	HSI FIB-4	Significant reduction in HSI (−2.36, *p* < 0.00001) and FIB-4 (−0.075, *p* < 0.016), related to the decrease of body weight, triglyceride levels, insulin resistance, and liver enzymes.
	Arai et al., 2022 [67]	Japan	16	-	Oral semaglutide daily 3 mg/4 weeks, then 7 mg/4 weeks, then 14 mg/16 weeks	24 weeks	CAP, LSM	Reduced HbA1c, HOMA-IR, ferritin normalized hepatic enzymes. Decreased FIB-4 index from 1.42 to 1.1. Significantly decreased CAP values from 344 to 279 dB/m.
	Ding et al., 2022 [68]	China	75/75	Metformin	Semaglutide sc OW 0.6 mg/1.2 mg/1.8 mg	12 weeks	US	Decreased AST, ALT, and GGT levels (*p* < 0.05). Significantly reduced moderate to severe NAFLD patients.
	Volpe et al., 2022 [69]	Italy	48	-	Semaglutide sc 0.25 mg OW for 4 weeks, then 0.5 mg OW for 20 weeks	52 weeks	US US-Liver Steatosis Score	Reduced body weight, HbA1c, HOMA-IR, serum lipid, AST, ALT, GGT, FLI. Significant improvement in liver steatosis severity, body composition.
	Flint et al., 2021 [70]	Germany	34/33	Placebo	Semaglutide sc 0.4 mg daily	72 weeks	MRE MRI-PDFF	≥30% reduction in liver fat content, but no significant change in liver stiffness. Decrease of HbA1c, liver enzymes, body weight.
	Newsome et al., 2021 [71]	United Kingdom	82/80	Placebo	Semaglutide sc 0.4 mg daily	72 weeks	Liver biopsy	Resolution of NASH (59% vs. 17% placebo, *p* < 0.001). Slowed fibrosis progression (4.9% vs. 18.8% placebo), but no significant reduction in fibrosis stages.
Dulaglutide	Kuchay MS, 2020 [72]	India	32/32	Standard of care without GLP-1RA	Dulaglutide 0.75 mg OW for 4 weeks, then 1.5 mg OW for 20 weeks	24 weeks	MRI-PDFF	Significant reduction of LFC (−3.5%, *p* = 0.025) and GGT levels (−13.1 IU/L, *p* = 0.025) in patients with NAFLD. No significant reductions in PFC, liver stiffness, serum AST and ALT levels.
	Bogomolov et al., 2020 [73]	Russia	65	-	Dulaglutide 0.75 mg OW for 2 weeks, then 1.5 mg OW for 24 weeks	26 weeks	FLI	Significant reduction of body weight, BMI, waist circumference, glucose, HbA1c, insulin resistance indexes, transaminases, and GGT. Decreased FLI and liver stiffness.

Abbreviations: GLP-1RA, glucagon-like peptide-1 receptor agonist; OW, once weekly; BID, twice a day; 1H-MRS, proton magnetic resonance spectroscopy; FIB-4, fibrosis-4, LDL-C, low-density lipoprotein cholesterol; ALT, alanine aminotransferase; AST, aspartate aminotransferase; GGT, gamma-glutamyl transferase; MRI, magnetic resonance imaging; VAT, visceral adipose tissue; SAT, subcutaneous adipose tissue; MRS, magnetic resonance spectroscopy; US, ultrasonography; LFC, liver fat content; T2DM, type 2 diabetes mellitus; VFA, visceral fat area; LAI, liver attenuation index; ACR, albumin-to-creatinine ratio; CRP, C-reactive protein; IL-1β, Interleukin-1β; BMI, body mass index; APRI, AST to platelet ratio index; NAFLD, nonalcoholic fatty liver disease; NFS, NAFLD fibrosis score; HSI, hepatic steatosis index; LSM, liver stiffness measurement, CAP, controlled attenuation parameter; NASH, nonalcoholic steatohepatitis; MRI-PDFF, magnetic resonance imaging-proton density fat fraction; IHCL, intra-hepatocellular lipid content; CT, computed tomography; MRE, magnetic resonance elastography; HOMA-IR, homeostatic model assessment for insulin resistance; FLI, fatty liver index; PFC, pancreatic fat content.

**Table 4 metabolites-13-00581-t004:** Ongoing trials investigating the effects of GLP-1RAs and next-generation-related molecules in patients with T2DM and NAFLD (www.clinicaltrial.gov, accessed 1 February 2023).

Trial Name	Estimated Enrollment	Start Date	Completion Date	Intervention	Criteria for Diabetes	Primary Outcome	Secondary Outcome
A Randomized, Double-Blind, Placebo-Controlled Phase 2 Study Comparing the Efficacy and Safety of Tirzepatide Versus Placebo in Patients with Nonalcoholic Steatohepatitis (SYNERGY-NASH) [112]	196 participants	19 November 2019	13 February 2024	Tirzepatide 5/10/15 mg sc OW vs. placebo	Patients with T2DM (HbA1c ≤ 9.5%) or without T2DM	Percentage of participants with absence of NASH with no worsening of fibrosis on liver histology.	≥1 point decrease in fibrosis stage with no worsening of NASH on liver histology. ≥1 point increase in fibrosis stage on liver histology. ≥2 point decrease in NAS on liver histology, with ≥1 point reduction in at least 2 NAS components. Change in liver fat content by MRI-PDFF. Change in body weight.
Researching an Effect of GLP-1 Agonist on Liver Steatosis (REALIST) [113]	93 participants	1 September 2019	30 March 2024	Dulaglutide 1.5 mg sc OW	Moderately controlled T2DM under OADs stable dose for at least 3 months	Histological improvement (regression of NASH) without worsening fibrosis over 52 weeks.	Changes in Fibrosis Kleiner score, Fibrotest score measurement, Fibrosis marker parameter, serum levels of ALT and AST, lipid profile, glycemic control, fat mass, quality of life, weight over 52 weeks, and changes in weight and AST and ALT levels over 24 weeks.
Combined Active Treatment in Type 2 Diabetes with NASH (COMBAT_T2_NASH) [114]	192 participants	26 March 2021	December 2023	Empagliflozin 10 mg/d + semaglutide 1 mg/week vs. empagliflozin 10 mg/d vs. placebo	Patient with T2D (HbA1c ≤ 9.5%) and NASH (F1–F3 fibrosis stage)	Histological resolution of NASH without worsening of fibrosis over 48 weeks.	Overall NAS, fibrosis stage, activity component of NASH, grade of steatosis over 48 weeks.
The Effect of Semaglutide in Subjects with Noncirrhotic Nonalcoholic Steatohepatitis (ESSENCE) [87]	1200 participants	1 April 2021	26 May 2028	Semaglutide sc OW	Patients with T2DM, without any glucose-lowering agents	Resolution of NASH and no worsening of liver fibrosis. Improvement in liver fibrosis with no worsening of NASH. Time to first liver-related clinical event.	Progression of liver fibrosis, NASH. Change in body weight, AST, ALT, lipid profile, histology-assessed liver collagen proportionate area, liver stiffness assessment.
A Phase IIb/III Randomized, Double-blind, Placebo-controlled Study to Evaluate the Safety and Efficacy of Cotadutide in Participants with Noncirrhotic Nonalcoholic Steatohepatitis with Fibrosis (PROXYMO-ADV) [115]	1860 participants	14 July 2022	26 March 2026	Cotadutide 300 μg/600 μg sc once daily vs. placebo	Patients with or without T2DM	Resolution of NASH without worsening of liver fibrosis based on biopsy.	Liver fibrosis improvement by at least one stage without worsening of NASH. Change in body weight, HbA1c, triglycerides. Progression to cirrhosis based on biopsy.
Effect on Nonalcoholic Fatty Liver Disease with Advanced Fibrosis in Patients with Type 2 Diabetes Mellitus on Treatment with Gastric Inhibitory Polypeptide/Glucagon-Like Peptide-1 Analogue NCT05751720 [116]	30 participants	April 2023	February 2025	Tirzepatide sc 0.25 mg OW/4 weeks then 0.5 mg OW, or Oral semaglutide 3 mg daily/first month, then 7 mg daily for 6 months	Patients with T2DM for > 1 year in presence of NAFLD advanced fibrosis	Change in liver stiffness in terms of kPa. Change in liver fat quantification.	Change in BMI. Glycemic control.
Semaglutide, 2.4 mg, Once Weekly: Effects on Beta-cell Preservation and Reduction of Intrahepatic Triglyceride Content in Obese Youth with Prediabetes (IGT)/Early Type 2 Diabetes (T2D) and Nonalcoholic Fatty Liver Disease (NAFLD) NCT05067621 [117]	60 participants	January 2023	January 2027	Semaglutide sc 0.24 mg/0.5 mg/1.0 mg/1.7 mg/2.4 mg OW	Impaired glucose tolerance or new onset T2DM (<6 months duration)	Change in oDI. Change in PDFF.	Change in oral glucose tolerance test (OGTT) derived biomarkers: oDI, fasting insulin, C-peptide. Time to glucose peak. Fractional rates of de novo lipogenesis. Changes in lipidic profile and liver fibrosis.

Abbreviations: T2DM, type 2 diabetes mellitus; NAFLD, nonalcoholic fatty liver disease; NASH, nonalcoholic steatohepatitis; GLP-1, glucagon-like peptide-1; NAS, NAFLD Activity Score; MRI-PDFF, magnetic resonance imaging—proton density fat fraction; oDI, Oral Disposition Index; OAD, oral antidiabetics; sc, subcutaneously; ALT, alanine aminotransferase; AST, aspartate aminotransferase; kPa, kilopascals; BMI, body mass index; IGT, impaired glucose tolerance.

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
