# Peer review of "Glucagon-like Peptide-1 Receptor Agonists in Patients with Type 2 Diabetes Mellitus and Nonalcoholic Fatty Liver Disease—Current Background, Hopes, and Perspectives"

_metabolites, 2023, doi:10.3390/metabo13050581_

Round 1
Reviewer 1 Report
The authors propose an extensive and comprehensive review of the litterature on GLP-1RA treatment of non-alcoholic fatty liver disease in type 2 diabetes. This review is well organized with 1) a complete introduction on the subject; 2) the effects of GLP-1RA and mechanism associated with its effects. The authors proposes several clear tables to gather 1) guidelines on GLP-1RA use, 2) studies on the effects of GLP-1RA and 3) ongoing trial on the effects of GLP-1RA.
It is a very clear and intersting review on this subject.
I have identified a few small changes to make :
-line 24 to 30 : this sentence is too long and should be split in several sentences
- Table 1 : you should separate each guideline by a horizontal line
- line 219 : tou should precise "displayed significant comparable weight loss"
- line 332 : replace 20,9% by 20.9%
Reviewer 2 Report
1 The title of the article fully reflects the content of the article.
Abstract briefly presents the problem of the lack of specific pharmacological methods for the treatment of non-alcoholic fatty liver disease (NAFLD). It has been suggested that glucagon-like peptide-1 (GLP-1 RAs) receptor agonists may act as potential candidates for the treatment of NAFLD in patients with T2DM. The purpose of the review is clear, the study is relevant and necessary for choosing a strategy for the treatment of NAFLD, complicating T2DM. All the information in Abstract is necessary, sufficient, reflects the essence of the article.
All Keywords are required.
In the "Introduction" section, the authors clearly showed the limited effectiveness of non-drug and drug therapy of NAFLD, which negatively affects patients with a combination of NAFLD and type 2 diabetes mellitus. In the section, the authors clearly identified the urgency of the problem: the need to find and develop effective and safe strategies for the treatment of NAFLD in patients with T2DM. The results of studies by independent research groups are presented, which show the possibility of influencing GLP-1RA on T2DM, obesity. The authors appropriately note that it is necessary to clarify the role of GLP-1RAs as a targeted therapy for NAFLD in people with or without diabetes.
The purpose of this review is clear. In this review, the authors identified and analyzed the benefits of GLP-1001RA therapy in the treatment of NAFLD complicating T2DM, updated the results of studies evaluating the effect of this hypoglycemic agent on fatty liver disease and fibrosis, and revised potential evidence-based recommendations.
In the section "Materials and methods", the authors indicated which terms were used to search for literature on PubMed and clinical trials. The authors examined the literature list of original studies, a review, systematic reviews and meta-analysis. The general idea of the study is clear.
In sections 3 and 4, the authors presented and analyzed in detail and clearly articles on the topics "Effects of GLP-1RAs on diabetes-related liver disease" and "Potential beneficial mechanisms underlying the effects of GLP-1RAs in diabetes-261 related liver disease", respectively. In section 5 "Perspectives", the authors showed promising agents (twincretin tirzepatide, cotadutide, liraglutide) for the treatment of T2DM with complications. All the presented results relate to the research topic, are presented in sufficient volume for their assessment, are described clearly, accompanied by a drawing and tables. Drawing and tables are necessary. The legends for the drawing and tables are clear.
The conclusions fully correspond to the results obtained. The authors are entitled to state that GLP-1 Rus may represent a powerful potential strategy for slowing the development of NAFLD in patients with T2DM.
The article is timely, does not cause any concerns. The manuscript did not cause any ethical problems. Statistical analysis corresponds to the study. All references to publications presented by the authors in the article are necessary and correct, made in the right style. Of the 107 links that are presented in the article, more than 76% have been in the last 5 years. I have no concerns about the similarity of this article with other articles published by the same authors.
Competing interests of authors do not create bias in the presentation of results and conclusions.
Reviewer 3 Report
Even though is a very interesting comprehensive review, there are some points that need clarification. It is suggested to review the design in detail, and present the most significant results and that the evaluation/analysis of these adhere to the main objective that led them to carry out this bibliographic review.
1. The manuscript needs writing and language editing. The title should be improved, for example, “Glucagon-like peptide 1 receptor agonists in patients with type 2 diabetes mellitus and nonalcoholic fatty liver disease”. The abstract should present the main point of this literature review. What is the purpose of this review? The main aim must be direct and the same throughout the manuscript (abstract, introduction, results/discussion) “to summarize the body of evidence supporting the effectiveness of GLP-1RA therapy in the management of T2DM complicated by NAFLD”. Authors should not use the words that appear in the title as keywords. References should be recent and relevant, they should be well referenced, and their use should be improved throughout the manuscript.
2. The introduction section should improve. A proper presentation and a good and clear justification (reason) for conducting this review study should be given. It would be better if the authors offered a hypothesis rather than the main objective of this study. It would be a good idea to rewrite the introduction section (between three to five paragraphs, going from general to specific), making known the main topics: the target diseases (NAFLD in T2DM), the effectiveness of the therapies (hypoglycemic agents, GLP -1AR) and end with the main hypothesis/objective (GLP-1AR as a treatment in patients with T2DM AND NAFLD) that the authors try to highlight with this review. Lines 101-103: This is part of the methodology. Line 106: Is this effect positive? It would be good to improve what the authors try to convey in figure 1. In Table 1: Is Semaglutide the GLP-1RA?
3. The methods section is sparse and needs to be improved. The description must be clear, concise, and detailed. The authors must declare that this review of the literature was carried out in accordance with the principles of the Declaration of Helsinki on data management. How was the review carried out (give details)? What results did the authors analyze? What were the inclusion and exclusion criteria? In which group of people was this review conducted (age, male, female, etc.)? What information was collected: treatment (dose), author, country, year, number of patients, type of study (details), study duration, diagnostic method, results, liver profile (enzymes), histopathological changes, etc.? All variables should be described, defined, and measured appropriately. In this section, the authors should make a brief description of the points to be analyzed regarding the results found in the different trials on the effect of GLP-1-RA on NAFLD in patients with T2DM.
4. It would be a good idea to add a results section to the manuscript to first describe the most important characteristics of the trials found, for example, the number of patients in total, which was the most used treatment, etc., summarizing the effects sought in tables 2 and 3. Lines 244-250: write it in the results section. How many articles did this search return? What were the most interesting and significant results? In Table 2: it would be a good idea to add a column with the diagnosis of the treated patients (DM2, obesity, NAFLD, NASH, etc.). Then, in the discussion section, analyze the effects found. It should be clear which were the most significant results collected from this literature review.
5. The discussion section should start with the main objective of this review study and the most significant results found. The first time that an abbreviation appears, the full name must be written. What does ESSENCE, SURPASS mean? The collected results (subsections) by the authors should be discussed from multiple angles and placed in context without overinterpreting them. Before the perspective subsection, it would be a good idea to summarize in a short paragraph what the authors found about the effects and potential mechanisms underlying GLP-1-RAs in T2DM-liver disease. If the authors hypothesize that GLP-1-RAs will be an effective treatment in patients with T2DM-NAFLD, this should be highlighted at the beginning of the perspective section. Given what the authors showed above and if the authors hypothesize that GLP-1-RAs will be an effective treatment in patients with T2DM-NAFLD, this should be highlighted at the beginning of the perspective section. A paragraph of limitations and suggestions for this review should be written before the conclusion.
6. The conclusion must be the same throughout the manuscript. The introduction, the study design, and the discussion of the results should lead the reader to the same conclusion as the authors.
I would like to encourage the authors to rewrite this review, thinking about the main objective of this study, and its design and responding with the results and arguments of the discussion to the most appropriate conclusion of this work.

Reviewer 4 Report
This review paper by Cazac and coworkers is very timely and well-written. The number of trials cited and referenced is excellent along with the perspectives given for the use of GLP-1RAs. This is especially true as Semaglutide is being used as a weight loss drug and demand is now higher than supply for diabetics, at least in the US. The content of the material in this manuscript is great. There are just a few items of suggested improvements.
1. Better spacing or horizontal lines in Table 1 would be helpful for readers.
2. A table of the GLP-1RA drugs summarizing their mode of action, dose range, side effects, route of admin, time on market, manufacturer, etc would also be helpful as not all readers are intimately familiar with all of these drugs.
Reviewer 5 Report
The Authors of the paper have reviewed the benefits of GLP-1RA therapy in the management of NAFLD that complicates T2DM. They have analyzed issues: effects of GLP-1RAs on diabetes-related liver disease, potential beneficial mechanisms underlying the effects of GLP-1RAs in diabetes-261 related liver disease, direct and indirect effects that promote NAFLD improvement. The aim of this comprehensive review was to identify and analyze the benefits of GLP-1RA therapy in the management of NAFLD that complicates T2DM, making an update on the studies that assessed the effects of this glucose-lowering agent on the fatty liver disease and fibrosis and revising the potential evidence-based recommendations. This paper is a some contribution to the scientific discussion about above issues.
Text and table editing and minor language revisions should be made. I recommend it for publication after revision.
Round 2
Reviewer 3 Report
The manuscript has been considerably improved. Given the nature of this review, the authors must make some changes to agree with the main objective of this review, which is to show the most relevant information to date on the benefits of GLP-1 RAS in patients with DM2-NAFLD.
Authors should not use the words that appear in the title as keywords.
Line 62: It would be good to add: GLP-1 RAs can be an effective treatment in patients with DM2-NAFLD, so the aim... It would be better to write “summarize” instead of “analyze”. Lines 26-30 and 63-66: Given the type of review, it would be better to write this: "describing the studies that evaluated the effects of these glucose-lowering agents in fatty liver disease and fibrosis, their possible mechanistic justification, current evidence-based recommendations, and next steps to be developed in the field of pharmacological innovation."
Line 139: Studies that did not include subjects with DM2 were excluded.
Line 393: Before conclusion: The increased appearance of NAFLD in patients with T2DM in a large proportion of patients, added to the still incomplete knowledge of the mechanical changes that these therapies induce in the liver, supports the need for more studies that seek effective treatments for both pathologies. A new contingent of drugs under development based on the principle of GLP-1 receptor agonism may represent a turning point in this field in the coming years.
Line 401-411: It would be better to write “Given the increasing worldwide incidence of both T2DM and NAFLD, this comprehensive review shows that in type 2 diabetic patients, GLP-1 RAs may contribute to the reduction of liver fat content, induce NASH resolution, and reduce the progression of fibrosis.”
